# A Preliminary Study of the Virome of the South American Free-Tailed Bats (*Tadarida brasiliensis*) and Identification of Two Novel Mammalian Viruses

**DOI:** 10.3390/v12040422

**Published:** 2020-04-09

**Authors:** Elisa M. Bolatti, Tomaž M. Zorec, María E. Montani, Lea Hošnjak, Diego Chouhy, Gastón Viarengo, Pablo E. Casal, Rubén M. Barquez, Mario Poljak, Adriana A. Giri

**Affiliations:** 1Grupo Virología Humana, Instituto de Biología Molecular y Celular de Rosario (CONICET), Suipacha 590, Rosario 2000, Argentina; bolatti@ibr-conicet.gov.ar (E.M.B.); chouhy@ibr-conicet.gov.ar (D.C.); viarengo@ibr-conicet.gov.ar (G.V.); 2Área Virología, Facultad de Ciencias Bioquímicas y Farmacéuticas, Universidad Nacional de Rosario, Suipacha 531, Rosario 2000, Argentina; pablocasal380@gmail.com; 3Institute of Microbiology and Immunology, Faculty of Medicine, University of Ljubljana, Zaloška 4, SI-1000 Ljubljana, Slovenia; Tomaz-Mark.Zorec@mf.uni-lj.si (T.M.Z.); lea.hosnjak@mf.uni-lj.si (L.H.); 4Museo Provincial de Ciencias Naturales “Dr. Ángel Gallardo”, San Lorenzo 1949, Rosario 2000, Argentina; euge_montani22@hotmail.com; 5Programa de Conservación de los Murciélagos de Argentina, Miguel Lillo 251, San Miguel de Tucumán 4000, Argentina; rubenbarquez@gmail.com; 6Programa de Investigaciones de Biodiversidad Argentina, Facultad de Ciencias Naturales e Instituto Miguel Lillo, Universidad Nacional de Tucumán, Miguel Lillo 205, San Miguel de Tucumán 4000, Argentina

**Keywords:** *Tadarida brasiliensis*, metagenomics, *Papillomaviridae*, TbraPV1, *Genomoviridae*, TbGkyV1

## Abstract

Bats provide important ecosystem services as pollinators, seed dispersers, and/or insect controllers, but they have also been found harboring different viruses with zoonotic potential. Virome studies in bats distributed in Asia, Africa, Europe, and North America have increased dramatically over the past decade, whereas information on viruses infecting South American species is scarce. We explored the virome of *Tadarida brasiliensis*, an insectivorous New World bat species inhabiting a maternity colony in Rosario (Argentina), by a metagenomic approach. The analysis of five pooled oral/anal swab samples indicated the presence of 43 different taxonomic viral families infecting a wide range of hosts. By conventional nucleic acid detection techniques and/or bioinformatics approaches, the genomes of two novel viruses were completely covered clustering into the *Papillomaviridae* (*Tadarida brasiliensis* papillomavirus type 1, TbraPV1) and *Genomoviridae* (*Tadarida brasiliensis* gemykibivirus 1, TbGkyV1) families. TbraPV1 is the first papillomavirus type identified in this host and the prototype of a novel genus. TbGkyV1 is the first genomovirus reported in New World bats and constitutes a new species within the genus *Gemykibivirus*. Our findings extend the knowledge about oral/anal viromes of a South American bat species and contribute to understand the evolution and genetic diversity of the novel characterized viruses.

## 1. Introduction

Bats belong to the order Chiroptera, which is the second-largest mammalian group, comprising 21 families and 1411 species distributed globally, with the exception of polar areas [1,2]. Approximately 25% of the world’s bat species are endangered, causing concerns about the negative conservation impact and its influence on the ecosystem services these bats provide, such as arthropod regulation, seed dispersal, and pollination [1,3]. On the other hand, certain specific aspects of bats—including their relatively long lifespan in relation to their body size [4], the reliance of some species on prolonged torpor [5], and flight—may make them suitable for hosting a wide variety of viruses [6], including zoonotic viruses highly pathogenic to humans [6], such as severe acute respiratory syndrome (SARS)-related coronavirus, Ebola virus, Nipah virus, and Hendra virus [7,8,9,10]. Nevertheless, little is known about their own pathogens [3]. In addition, the gregarious behavior of many bat species, such as free-tailed bats *Tadarida brasiliensis* (I. Geoffroy Saint-Hilaire, 1824), may facilitate rapid transmission of pathogens between bats and other species [6]. *T. brasiliensis* is the most abundant migratory and cosmopolitan species of the New World bats, widespread throughout the Americas [11,12,13] and protected by international agreements [14].

Using next-generation sequencing (NGS) technologies, an enormous variety of viral species and genotypes [15,16] have been identified in the tissues and feces of bats mainly inhabiting Asia [17,18], Africa [19,20], Europe [21,22], and North America [23,24]. On the other hand, the viromes of bats from South America remain understudied [25,26]. For example, the identification of viruses infecting *T. brasiliensis* is principally limited to detection of specific viral families, such as rabies lyssavirus [27], alphacoronaviruses [28], polyomaviruses [29], circoviruses [30], and anelloviruses [31].

In order to contribute to the preservation of *T. brasiliensis* and to evaluate its possible role as a pathogen reservoir, greater efforts directed at identifying the viruses present in this species are needed. In this study we report a detailed description of two novel complete genome sequences, one describing a new papillomavirus genus and the other representing a novel variant of an existing gemykibivirus species. In addition, we report a preliminary overview of the *T. brasiliensis* virome composition. Altogether, our findings add to the knowledge of viral diversity in a South American bat species, providing insights for understanding their role as reservoirs, as well as their own pathogens, which may have consequences for the animals’ health.

## 2. Materials and Methods

### 2.1. Study Area, Sample Collection, and Ethics Statement

The bat colony investigated occupies the attic of the Law School building at the Universidad Nacional de Rosario in downtown Rosario, Argentina (32°56′36.76″ S 60°39′02.09″ W) [32]. In this place, *T. brasiliensis* (*Molossidae*) establishes a maternity colony every year that can reach about 30,000 individuals during the maternity season (November to March), after which they migrate [32,33].

A total of 98 swab samples (49 oral and 49 anal) were collected from 49 adult female specimens inhabiting this colony from December 2015 to February 2016. Briefly, bats were manually captured from the walls and held in individual cotton bags for determination of their species based on anatomical and morphological characters, reproductive condition, and age [33].

The oral cavity and anal regions of each individual were sampled using individual sterile cotton-tipped swabs (Deltalab, Barcelona, Spain), rolled back and forth (10 times), suspended in 200 µL of saline solution (NaCl 0.9%), and stored at 4 °C until further processing. The bats were rehydrated and released.

During this study, every effort was made to minimize interference with and suffering of the animals; no breeding or pregnant females were captured, and no animals were sacrificed. Sample collection was conducted by trained professionals as approved by the Ministry of Environment of the Argentinian Santa Fe Province (file 021010016257-1) and Facultad de Ciencias Bioquímicas y Farmacéuticas (Universidad Nacional de Rosario) animal ethics committee (file 6060/243, 20 March 2015).

### 2.2. Sample Processing and Viral DNA Enrichment

Samples were processed according to previously published protocols that have been successfully applied for identification of papillomavirus (PV) in human skin swab samples [34,35,36]. Briefly, the cells were centrifuged at 13,000× *g* for 5 min and the pellets were resuspended in 100 μL TE buffer (Qiagen, Hilden, Germany) containing 100 μg of proteinase K (Qiagen), and incubated overnight at 55 °C. Following proteinase K inactivation (95 °C for 10 min), the lysates were stored at −20 °C. Subsequently, the obtained samples were tested for the presence of PV DNA using improved versions of FAP [37,38] and CUT PCRs [35], as described previously [36,39]. Circular DNA molecules in lysates of five selected PV-positive samples (four anal and one oral swab) were enriched using rolling-circle amplification (RCA) with the illustra TempliPhi 100 Amplification Kit (GE Healthcare, Chicago, IL, USA) [40,41,42].

### 2.3. Next-Generation Sequencing (NGS), Read Quality Control, and Sequence Filtering

The pool of RCA-enriched samples was sequenced on an Illumina Hiseq 4000 instrument at the sequencing facility of GATC Biotech (Ebersberg, Germany). Sequencing libraries were prepared using the GATC automatic library preparation approach, and the sequencing reads were sequestered in the format of 2 × 150 bp. Reads were subjected to quality trimming and filtering using the bbduk program (BBTools v38.42). End trimming was performed on the first and last 15 bases of each read, clipping bases with PHRED scores below 15. Trimmed reads shorter than 120 bp and with an average PHRED score below 20 were discarded.

The read pairs contained in the metagenomic sample, which shared k-mers, sliding-window subsequences of 27 nt, with the sequencing datasets of samples (six in total) that were processed and analyzed in the same sequencing batch, were discarded using the bbduk program (referred to as laboratory-batch background screen in Figure 1). The primary purpose of this step was to conservatively limit the possibility of falsely identifying viral taxa that did not originate from the bat metagenomics sample and that could have been introduced by aerosol during sample processing or index hopping during sequencing.

In order to limit the content of bacterial reads, the metagenomic dataset was mapped to the bacterial reference-index files (obtained 6 November 2017, from ftp://ftp.ccb.jhu.edu/pub/infphilo/centrifuge/data/p_compressed.tar.gz) using the centrifuge sequence classification system (Centrifuge version 1.0.3-beta) [43]. Reads not mapping to any bacterial taxon were used in further metagenomic analyses (unless stated otherwise).

### 2.4. Metagenomic Analysis Workflow

Two types of metagenomic characterization workflows were used: (1) taxonomic classification of NGS read pairs and (2) taxonomic classification of contigs assembled *de novo* from NGS read pairs. In both cases, the centrifuge metagenomic classification system with the reference nucleic sequence index files, obtained from ftp://ftp.ccb.jhu.edu/pub/infphilo/centrifuge/data/p_compressed+h+v.tar.gz (version 12 June 2016, downloaded 6 November 2017), was used to obtain the final taxonomic calls (default parameter settings). Taxonomic classification of sequences was further summarized to the taxonomic level of family using pavian [44].

*De novo* assembly was performed with two different De Brujin graph assembly tools: SPAdes (v3.11) [45] and Unicycler (obtained from github: 27 October 2017; github commit: 220d5daebc8267d37378f191e14acb5c5a1ff757), adapting various parameter settings. Altogether, six different metagenomic *de novo* assemblies were constructed, using settings specified in Appendix A. All contigs assembled *de novo* by any of the six approaches exhibiting a minimum length of 500 nt were collected and subjected to taxonomic classification (workflow 2).

### 2.5. Characterization of Novel Viral Genome Sequences and Phylogenetic Analysis

The circularity of the complete genome assemblies was determined by matching the sequence stretches (minimum match length 50 nt) at their 5′ and 3′ ends. Coverage statistics of the novel complete viral genome sequences were obtained by remapping the trimmed read dataset to the constructed genome assemblies using bowtie2 (v2.2.6) [46].

Sequences of the *E1*, *E2*, *L2*, and *L1* genes of 376 reference PV genomes, downloaded from PaVe (http://pave.niaid.nih.gov/ on 13 March 2019), and the corresponding genes from the novel PV, *Tadarida brasiliensis* papillomavirus type 1 (TbraPV1), were used in the phylogenetic analysis. Additional information on nucleotide sequences (GenBank accession number and virus name abbreviations) used in these analyses is summarized in Appendix A. The *E1E2L2L1* concatenation was constructed by first obtaining the amino acid-guided multiple sequence alignments of each gene. Multiple sequence alignments were obtained using Muscle (v3.8.31) [47]. As suggested by Bernard et al. (2010) [48], the multiple sequence alignments used for phylogenetic analysis of TbraPV1 were guided by the amino acid alignments and the PV identity calculation was based on patristic distance measurements, as determined by SeaView [49]. Phylogenetic clustering was conducted using IQ-TREE [50]. The most appropriate substitution models were determined using ModelFinder [51], according to the Bayesian information criterion. Branch support values were calculated using UF bootstrap (1000 replicates) [52], SH-aLRT (1000 replicates), and aBayes tests [53].

Phylogenetic analysis of *Genomoviridae* was conducted using a reference dataset of 166 complete genome nucleotide sequences and 166 Rep protein sequences downloaded from NCBI GenBank (19 March 2019). The complete genome sequences were rotated to all begin in the start codon of the *Rep* gene using circulator (version: GitHub commit a4befb8c9dbbcd4b3ad1899a95aa3e689d58b638) [54], and the two subunits of the *Rep* gene were concatenated into a single protein sequence in which the GenBank record indicated them as parts of different genes/coding sequences. Pairwise sequence identity values used for taxonomic classification of *Tadarida brasiliensis* gemykibivirus 1 (TbGkyV1) were obtained using Sequence Demarcation Toolkit (SDT v1.0) [55]. In this scope, the pairwise sequence alignments were produced using Muscle (v3.8.1) [47]. Phylogenetic trees were rendered using FigTree (v1.4.4) (http://tree.bio.ed.ac.uk/software/figtree/), and sequence identity histograms were visualized using gnuplot (v1.5).

Open reading frames (ORFs) of novel viral genome sequences were marked using ORFfinder (NCBI); the manual annotation process of the ORFs was guided by the use of NCBI BLASTp, and the identification of viral-family specific sequence motifs was performed using regular expressions with the linux grep utility (v2.16). Members of *Papillomaviridae* and *Genomoviridae* identified in bat species so far are summarized in Appendix A.

### 2.6. Complete Papillomavirus Genome Confirmation

The complete genome sequence of the novel PV type (TbraPV1) was obtained by generating four overlapping amplicons in different PCR reactions, using 1 X PCR buffer, 3.5 mM of MgCl_2_, 200 µM of each dNTP (Thermo Fisher, Walthem, MA, USA), 1.25 U of GoTaq HotStart polymerase (Promega, Madison, WI, USA), and 0.8 µM of each of the primers (TbraPV1-1F 5′-cagggtattcagggtgtttctcc-3′ and TbraPV1-1R 5′-aatgtttctaatctgcaacc-3′; TbraPV1-2F 5′-gtgcgcggcgacttctcatactta-3′ and TbraPV1-2R 5′-tcagcctcattgtcctcatcattg-3′; TbraPV1-3F 5′-tgggcttgaaacctggacactaca-3′ and TbraPV1-3R 5′-atgcccgggaatatggatgga-3′; TbraPV1-4F 5′-ggcctgcaagaccacctac-3′ and TbraPV1-4R 5′-gggggcatctgacctgtta-3′). Cycling conditions for the four reactions were the same and were performed as follows: initial denaturation at 95 °C for 2 min, followed by 45 cycles of 40 s at 94 °C, 40 s at 50 °C, and 2 min at 72 °C, with a final extension at 72 °C for 5 min. The amplicons were resolved in a 1% agarose gel electrophoresis and the ~2 kb fragments were gel purified, ligated into the pGem-T Easy Vector (Promega), and transformed into *E. coli* cells. Sanger sequencing was performed using sequencing facilities at Macrogen Inc. (Seoul, Korea). In August 2019, DNA clones and the corresponding nucleotide sequences were subsequently submitted to the Animal Papillomavirus Reference Center (http://www.animalpv.org/) for its confirmation and official designation.

### 2.7. Nucleotide Sequence Accession Number

The GenBank/EMBL/DDBJ accession numbers for the novel viruses reported in this paper are TbraPV1 (MN329804) and TbGkyV1 (MN329805). The relevant raw high throughput sequencing data obtained in this study was deposited at the NCBI Sequence Read Archives (SRA) with the following accession number: PRJNA615356. The contigs, obtained by *de novo* assembly as part of the metagenomic workflow (2), have also been made available for download (Appendix A).

## 3. Results

### 3.1. Global Analysis of High-Throughput Sequencing Data

NGS data analysis workflows and centrifuge-based taxonomic assignments of reads and contigs are depicted in Figure 1 and Table 1, respectively. Briefly, a total of 10,409,798 read pairs were sequestered from the RCA-enriched samples, and 10,220,118 of them passed the quality filtering and trimming procedures. Out of these, 6,738,566 read pairs were removed during the laboratory-batch background screen and an additional 878,852 read pairs were identified as originating from bacteria. The final metagenomic characterization was carried out using the remaining 2,602,700 read pairs (Figure 1). Metagenomic analysis revealed that only a small proportion of read pairs (13,897 read pairs; 0.534%) and *de novo* assembled contigs longer than 500 nt (153 out of total 42,891; 0.357%) mapped to viral taxa.

Overall, a large number of phage-related sequences were detected (77.3% of viral read pairs and 39.9% of viral contigs), likely representing the most abundant entities infecting bacteria present in the bat digestive system, which exhibited similarity mostly to the families *Inoviridae*, *Siphoviridae*, and *Myoviridae* (Table 1). The eukaryotic viral sequences (insect, invertebrate, plant, protist, and vertebrate viruses) could be summarized into a total of 35 viral families, 22 corresponding to viral families with DNA genomes, and 13 to families with RNA genomes.

Sequences of 10 different viral families infecting insects and crustaceans, mostly found related to the families *Baculoviridae*, *Ascoviridae*, *Iridoviridae*, and *Nimaviridae*, were detected (0.900% of viral read pairs and 12.4% of assembled viral contigs). On the other hand, sequences related to viruses infecting plants (five viral families, 0.446% of viral read pairs, and 4.57% of viral contigs) were mostly associated with *Phycodnaviridae* or *Potyviridae,* whereas those related to viruses infecting protists (five viral families, 13.6% of viral read pairs, 0.654% of viral contigs) clustered predominantly in the family *Mimiviridae*.

Sequences classified as originating from vertebrates, predominantly mammalian viruses, were represented by 7.07% of viral read pairs and 35.9% of viral contigs. The principal viral families identified included *Retroviridae*, *Genomoviridae*, *Herpesviridae*, *Papillomaviridae*, and *Poxviridae*. The analysis also identified (although in low counts) viral sequences related to the family *Alloherpesviridae*, which infects fish and amphibians. Of note, the metagenomic analysis indicated that 205 out of 461 read pairs were assigned to the family *Retroviridae* (1.48% of viral read pairs), and 22 assembled contigs (14.4% of viral contigs; Table 1) exhibited resemblance to the nucleotide sequence of *Desmodus rotundus* endogenous retrovirus isolate 824 (Genbank accession number: NC_027117) [56]. However, a more detailed analysis of this sequence revealed the presence of two flanking regions at the 5′ and 3′ ends of approximately 1000 nt, which probably derived from the host genome (*Desmodus rotundus*). In fact, contigs and reads previously classified as *Retroviridae* in our study aligned with these flanking regions. This finding explained the initial misclassification and indicated that great care should be taken when using GenBank sequence data as reference material because a large portion of sequences and their respective annotations may not have been curated adequately.

Finally, a total of 96 read pairs and 10 assembled contigs were classified as similar to viruses unassigned to taxonomical families (Appendix A).

### 3.2. Identification of Novel Bat Viruses

#### 3.2.1. Papillomaviridae

The metagenomic analysis suggested that a total of 90 read pairs (workflow 1) and 14 contigs (workflow 2) could be attributed to PVs (Table 1). The longest contig obtained by assembly *de novo* covered the complete genome of TbraPV1, which was subsequently confirmed by conventional molecular methods (PCR, cloning, and Sanger sequencing; data not shown).

Remapping read pairs (quality-, background-, and bacteria-filtered read pairs) to the confirmed TbraPV1 genome sequence indicated a complete sequence length of 8151 nt, with a GC content of 46%. The complete novel genome sequence was covered on average 252.6× by a total of 6935 read pairs (13,870 reads). Detailed analysis of the TbraPV1 viral genome (Table 2 and Figure 2A), showed a typical genomic organization of bat PVs, potentially encoding four early genes (*E6*, *E7*, *E1*, and *E2*) and two late genes (*L2* and *L1*) [57,58]. A putative *E4* gene was found overlapping the E2 gene (Figure 2A), with its own start and stop codons, and the presence of E4-characteristic proline-rich stretches (12.7%) with an important role in cell cycle arrest was found [59].

The upstream regulatory region (URR) of TbraPV1 contained two typical TATA boxes, three putative E2 protein binding sites, an E1 protein binding site [60], and three putative polyadenilation sites for late gene transcripts (Table 2). Multiple potential binding sites for transcriptional regulatory factors, such as AP-1, NF-1, and SP-1, were also present within the URR (data not shown).

Typical domains were additionally identified in the putative viral proteins encoded by TbraPV1. The E6 protein contained two characteristic zinc-binding domains, separated by 36 amino acids [61], and four internal and likely not functional PDZ-binding motifs (RTNV, ISDL, SSIL, LSSL) [62]. The E7 protein contained a pRB-binding motif (LWCDE) [63] and a single zinc-binding domain (Table 2). Analysis of the E1 protein, the largest protein encoded by TbraPV1 (Table 2), showed the typical ATP-binding site of the ATP-dependent helicase (GPSNSGKS) [64], and several Cdk-phosphorylation and Cyclin-binding sites. A highly conserved bipartite-like nuclear localization signal (NLS) and a leucine-rich Crm1-dependant nuclear export signal (NES) (LSPVLEKVTI), which together allow shuttling of the E1 protein between the cell nucleus and the cytoplasm in most human PVs [65,66,67], were identified at the N-termini of the E1 protein. No conserved leucine zipper domain was present at the C-termini of the putative TbraPV1 E2 protein, in agreement with other bat PVs (EsPV2 and RfPV1) [58]. At the N-termini of the L2 protein, a highly conserved furine cleavage motif (RRKR), as well as a transmembrane-like domain (GTGGGGRGVPIGPRVATGRPGGPINSVG) [68], were identified. In addition, a canonical polyadenylation site, necessary for regulation of early viral transcripts [69], was also found in the TbraPV1 *L2* gene (Table 2).

Phylogenetic analysis, based on 377 PV *L1* gene nucleotide sequences, indicated peak sequence identities of TbraPV1 to HPV41 and RfPV1, which amounted to 61.5 and 60.6%, respectively (Figure 3, charts A2 and A4). Maximum likelihood phylogenetic clustering of *L1* sequences (Figure 4) suggested common ancestry of TbraPV1, EdPV1, and HPV41, and that TbraPV1 branched away prior to the delineation of EdPV1 and HPV41, with high SH-aLRT, aBayes, and UF bootstrap support values. Further phylogenetic analyses were conducted based on the concatenated alignments of 377 *E1*, *E2*, *L2*, and *L1* gene nucleotide sequences, and they indicated a peak sequence identity of TbraPV1 to RfPV1 (54.1%, Figure 3, charts A1 and A3), whereas the maximum likelihood phylogenetic clustering indicated analogous common ancestry to the concatenated PV genes (Figure 5), with high SH-aLRT, aBayes, and UF bootstrap support values.

#### 3.2.2. Genomoviridae

The metagenomic analysis indicated the presence of several genomovirus-related read pairs and sequence contigs (Table 1), and the complete genome sequence (2196 nt) of TbGkyV1 was recovered using *de novo* assembly. Remapping to the TbGkyV1 genome sequence indicated a mean coverage of 59.4× by a total of 430 read pairs (860 paired reads + 7 unpaired reads) and did not reveal any abnormalities that would indicate misassembly. Completeness of the circular genome sequence was determined by matching 5′ and 3′ ends, and the sequence was rotated to begin with the characteristic *Genomoviridae* nonanucleotide motif [70]. Three non-overlapping ORFs, with a minimum protein length of 120 aa, were found, exhibiting peak amino acid similarities to the *Genomoviridae* Rep and CP/Cap proteins. The Rep protein of TbGkyV1 was encoded across two different Rep-encoding ORFs separated by an intron, representing a catalytic and a central protein domain (Figure 2B, Table 3), which is characteristic for replication-associated proteins encoded by single-stranded (CRESS) DNA viruses [71].

Phylogenetic analysis and pairwise sequence comparison demonstrated that TbGkyV1 shares its highest identity in the Rep protein (77.5%) and the complete genome level (nucleotide sequence, 77.2%) with the “Mongoose associated gemykibivirus 1” (Genbank accession number KP263545) [70] (Figure 3B and Figure 6).

Genomic features and conserved domains of the novel gemykibivirus are shown in Table 3 and Figure 2B. The large intergenic region of the novel virus contains the characteristic nonanucleotide (TATAAATAG) motif, which is likely to be important for rolling-circle replication initiation [72,73,74]. The Rep protein’s catalytic domain of TbGkyV1 contained rolling circle replication motif I (LFTYSQ), possibly involved in the recognition of iterative DNA sequences associated with the origin of replication [70], motif II (THLHV), which may regulate the nicking/joining endonuclease activity at the origin of DNA replication [73,75], motif III (YATK), involved in the double-strand DNA cleavage [73,76], and a geminivirus Rep protein sequence (GRS) (RLFDVENFHPNIVPSR), which allows appropriate spatial arrangements of motifs II and III [77]. Furthermore, Rep helicase motifs Walker-A (GPSRTGKT), Walker-B (VFDDI), and Walker-C (WLMN) [78,79,80,81] were identified in the central Rep protein domain. Walker motifs contribute to ATP binding, which is used as an energy source to unwind the dsDNA intermediate in the 3′–5′ direction by the Rep helicase [80,81].

## 4. Discussion

More than 200 viruses from 27 taxonomic families have been isolated from or detected in bats so far [16], with a few of them implicated in the etiology of several severe diseases in humans. Except for rabies, no direct evidence of zoonotic diseases transmitted by New World bats has been found [27]. New World, especially South American, and Old World bat species had different evolutionary histories, leading to distinct immunological features [3,16]. Viruses infecting South American bat species have been poorly studied [24,26], and further research focused on evaluating their viromes is required. Here, a first attempt to assess to the virome composition in pooled oral and anal swabs of *T. brasiliensis* was presented.

During our study a total of 6,738,566 read pairs were removed during the laboratory batch-background screening, due to traces of sequences from co-processed samples. The power of NGS stems from non-specific sampling of nucleic acids and automated repetition, yielding vast numbers of sequencing reads, providing the opportunity to characterize populations of nucleic acids with unprecedented sensitivity, accuracy, and non-specificity. Due to its super-sensitivity, even the slightest addition of environmental nucleic acids to a sample may be detected using NGS and can potentially further complicate the interpretation of the results. Laboratory background and/or DNA isolation kit-derived contamination has been addressed previously as a major factor that can severely impede the interpretation of high throughput sequencing data, and the use of negative controls has been proposed [82]. Moreover, positive/negative control samples have been recommended in metagenomic experiments aimed at detecting pathogens in clinical samples [83]. In this study, the metagenomic sample was processed alongside six samples of molluscum contagiosum skin lesions, and laboratory background filtering was initially considered due to the detection of approximately 90 molluscum contagiosum virus read pairs in the trimmed read dataset. In order to prevent the identification of human skin microflora in the pooled bat swab sample, a strict k-mer based negative filtering was used, effectively removing any read pair that contained at least one 27 bp subsequence that could be identified in any of the read pairs from the six background datasets. Because the background samples also originated from mammalian (human) skin, it could be that a large portion of mammalian reads were removed during this step, explaining the somewhat extreme number of read pairs classified as laboratory-batch background. Moreover, it is also likely that the viral composition of human and bat skin could be shared to some degree, but due to the filtering scheme reads originating from bats’ anal and oral microflora sharing nucleotide sequence similarity to that of human skin may have also been removed prior to metagenomic classification. As a consequence, taken strictly, only the subset of viruses that are present in *T. brasiliensis,* but not in human skin, was explored here. Although DNA spillovers could be suspected and controlled for to some degree, in this case it may be a greater challenge in studies where multiple samples from different species or anatomical sites of bats are processed. In light of the present results, it would likely be beneficial to process the samples in such studies as independently as possible and to include negative controls that would characterize the laboratory background, such as sequencing libraries of buffer solutions that underwent the same treatment as the samples, as suggested previously [84,85].

Specifically, a total of 13,897 virus-related read pairs (0.53% out of 2,602,700) and 153 virus-related contigs (0.36% out of 42,891) were assembled *de novo*, mapped to viral taxa, and identified by NGS. Although the proportion (and number) of virus-related sequences detected in this study (<1%) is comparable to reports of previous studies of bat viromes based on Illumina sequencing [19,21,86], it may be that additional physical viral DNA/RNA enrichment steps, such as centrifugation, filtration, and/or nuclease-treatment, could further augment the viral read yields, as suggested previously [87]. Initially, this study was focused on the identification of PVs in *T. brasiliensis,* in order to explore their diversity in different hosts. Accordingly, swab samples included in this work were first processed using experimental protocols, designed previously to suit our aforementioned initial aim [34,35,36]. Viral DNA was enriched using RCA, as suggested previously by others [40,41,42]. However, it should be noted that RCA may have favorably facilitated the amplification of circular genomes and, as a consequence, hindered the detection of linear genomes. Thus, more than 80% of the classified viral sequences (11,162/13,801) identified in our analysis corresponded to circular viral genomes.

In this study about 4.5% of viral reads and 60% of contigs corresponded to sequences from 35 eukaryotic viral families, mostly with DNA genomes. Interestingly, virus-associated sequences from RNA viruses belonging to 14 families were also detected, most likely reflecting the presence of traces of viral RNA. Limited but highly accurate reverse transcriptase activity has indeed previously been reported for the phi29 DNA polymerase, used in RCA [88].

Most of the insect-infecting viral sequences detected belonged to viral families infecting lepidopteran adults or larvae, which may represent the diet of *T. brasiliensis*, as detected previously in feces and anal swabs from various insectivorous bat species (*Myotis* sp., *Rhinolophus* sp., *Molossus* sp., *Neoromicia* sp.), including *T. brasiliensis* [19,21,23,26]. In addition, the detection of various plant viral families in this study could reflect the plant diet of the insects ingested by the bats.

A total of 15 different mammalian viral families were identified in *T. brasiliensis* samples, representing approximately 43% (15/35) of the eukaryotic viral families interrogated herein. Several mammalian viral families, supported by the contigs and sequencing reads, have been identified previously in New World [23,24,26] and Old World [17,18,89] bat species. The mammalian viral families identified in *T. brasiliensis* included typical zoonotic viruses identified previously in bats, such as *Polyomaviridae* [29], *Rhabdoviridae* [90], *Coronaviridae* [23,28], *Poxviridae*, *Flaviviridae*, and *Adenoviridae* [23]. The identification of *Circoviridae* and *Astroviridae* in *T. brasiliensis* was also in line with the results of previous studies [23,31]. On the other hand, this study indicated the presence of *Genomoviridae, Alloherpesviridae*, *Papillomaviridae*, *Herpesviridae*, *Paramyxoviridae*, and *Reoviridae* in *T. brasiliensis* for the first time. Notably, the presence of incorrect annotations in public databases, such as the sequences assigned to the *Retroviridae* family in this study, highlight the need for the curation of data (whenever possible) to avoid the under- and/or overestimation of the classified sequences derived from metagenomics studies.

PVs are a highly diverse family of non-enveloped and double-stranded circular DNA viruses that are known to infect a wide variety of mammals, as well as birds, reptiles, and fish [91,92]. Various human and non-human PVs, including bat PVs, have frequently been identified in healthy epithelia and may represent part of the native epithelial microflora [34,57,58]. Several studies have suggested the presence of PVs in Old World bat species using conventional [57,58,93] or NGS aproches [20,41,94,95]. The only PV type (MmoPV1) identified in a New World bat species (*Molossus molossus*) has recently been described [26], suggesting a crude sampling imbalance and a severe lack of information to elucidate the evolutionary mechanisms driving PV diversification on the global scale. In this study, TbraPV1 has been identified in pooled oral and anal swabs of *T. brasiliensis* by NGS, and its sequence has been completely characterized by conventional molecular techniques. TbraPV1 is the first reported PV type found in *T. brasiliensis* and the second PV type identified in New World bat species.

According to the current ICTV *Papillomaviridae* classification guidelines (published in June 2018), based on the nucleotide sequence of the *L1* gene [96], TbraPV1 is the founding member of a novel PV genus in the *Firstpapillomavirinae* taxonomical subfamily, sharing more than 45% sequence identity to other PV types included in this subfamily. Although nucleotide sequences analysis in the *L1* gene indicated that TbraPV1 shares a 61.5% identity with HPV41 (*Nu*-PV) and should be included within the same genus (more than 60% of nucleotide identity in *L1* gene) [91], the mentioned demarcation criteria suggests a visual inspection of phylogenetic trees derived from concatenated *E1, E2, L1*, and *L2* nucleotide sequences to delineate PV genera [96]. In the present study, such analysis clustered TbraPV1 basal to the delineation of HPV41 (*Nu-*PV) and EdPV1 (*Sigma-*PV), identified in a North American porcupine (*Erethizon dorsatum*) and, therefore, may represent a novel genus within the *Papillomaviridae* family. This phylogenetic clustering also indicated that TbraPV1 shares common ancestry with other bat PVs such as EsPV1, EsPV3, RfPV1, EhPV1, and MscPV2. On the other hand, TbraPV1 is only distantly related to RaPV1, EhPV2, EhPV3, EsPV2, MscPV1, and MrPV1, which have been identified from different tissues and bats species. The idea that different bat PVs evolved during a process of strict host coevolution is further refuted by the observation that different bat PVs appear scattered around the *Papillomaviridae* phylogenetic tree in a highly polyphyletic manner [57,58]. In addition, under strict host coevolution it would be expected for TbraPV1 and MmoPV1, both molossid PVs, to be closely related; nevertheless, MmoPV1 has a basal taxonomic position with respect to TbraPV1. These observations support the idea of multiple evolutionary forces as drivers of PV evolution, including coevolution, adaptive radiation, broad host range, host switch, and recombination [58].

Genomoviruses are single-stranded circular DNA viruses that belong to the recently proposed *Genomoviridae* family [97]. Members of this family encode two genes—the *Cap/CP* and the rolling-circle replication-associated protein (*Rep*)—and an intergenic region. It has been proposed that a novel viral complete genome sequence of the same species exhibits more than 78% similarity to any other complete genomovirus genome [70]. In addition, in previous studies, the authors aimed to establish nine genera within the family *Genomoviridae* based on pairwise comparisons of complete genome sequences [70]. CRESS DNA viruses, including genomoviruses, have been found in association with a great variety of animal species, such as camels [98], bats [89], mongooses, badgers [99], wolves [100], pigs [101], and humans [102], as well as in environmental-associated [103] and plant-associated [104] samples. However, no direct implication with a disease has been demonstrated so far. In particular, bat-associated genomoviruses have been identified from feces [105,106] or pharyngeal and anal swab samples [89] of Asiatic [89,105] or European [106] bat species and have been attributed to various taxonomic genera of the *Genomoviridae* family [70]. TbGkyV1 is a novel species within the *Gemykibivirus* genus according to the classification criteria [70]. It should be noted that the Rep and Cap proteins of TbGkyV1 exhibited different percentages of similarity, the Cap being considerably more divergent than the Rep, indicating differences in their evolutionary histories due to their respective molecular functions [70,106]. To the best of our knowledge, this is the first report demonstrating the presence of genomoviral sequences in mucosal swab samples of a New World bat species.

Finally, it is worth noting that the results of our metagenomic screening of the pooled *T. brasiliensis* oral and anal swab samples is effectively provided at three different levels of specificity/sensitivity. The two complete genome sequences (TbraPV1 and TbGkyV1), that have been described at the highest level of detail, also confer the highest level of confidence. In other words, we have complete confidence that these two viruses were present in the pooled nucleic acid samples.

Assigning sequences that did not assemble at the level of complete viral genomes to taxonomical families could therefore be a valid approach offering a higher level of sensitivity than only the complete genome assemblies, but at the cost of diminished specificity. In addition, identifying a sequence fragment that resembles a known viral genome in a given genomic region, may not always be sufficient to infer that that specific virus was present in the sample. Viruses are highly promiscuous entities, which can easily exchange parts of their genomes with their hosts, be integrated or even naturalized into the host genomes [107]. Taxonomical viral families identified among the assembled contigs could be interpreted as viral families that were probably represented in our samples. Lastly, and due to the low number of assembled contigs, that were found related to known viral sequences, we attempted to increase the sensitivity even further by obtaining taxonomical family mappings also for the source read pairs. These results, however, should be interpreted with utmost caution, because they likely confer a very low level of specificity due to the limited sequence length (2 × 150 bp). Taxonomical viral families identified by read-pair taxonomy mapping only, merely suggest the possibility that these families were present and should be replicated in the future by taxonomical mappings conferring greater specificity, for example with longer sequences, such as those assembled from Illumina or Nanopore reads. 

## 5. Conclusions

This study presents an initial description of the oral/anal virome composition of *T. brasiliensis*, a widely-distributed New World bat species living in close contact with the human population, for the first time. Although their biological significance is not clear, this work contributes to a better understanding of the evolution and genetic diversity of these viruses. Using conventional nucleic acid detection techniques and/or bioinformatics approaches, the whole genomes of two novel viruses were completely covered, TbraPV1 and TbGkyV1, clustering into the *Papillomaviridae* and *Genomoviridae* families. TbraPV1 is the first PV type identified in this host and the prototype of a novel genus in the *Firstpapillomavirinae* taxonomic subfamily. TbGkyV1 is the first genomovirus reported in New World bats and constitutes a new species within the genus *Gemykibivirus*. Future studies are required to investigate the possible health impact of the viruses described on bat colonies and to identify the factors that contribute to their dispersal.

## Figures and Tables

**Figure 1 viruses-12-00422-f001:**
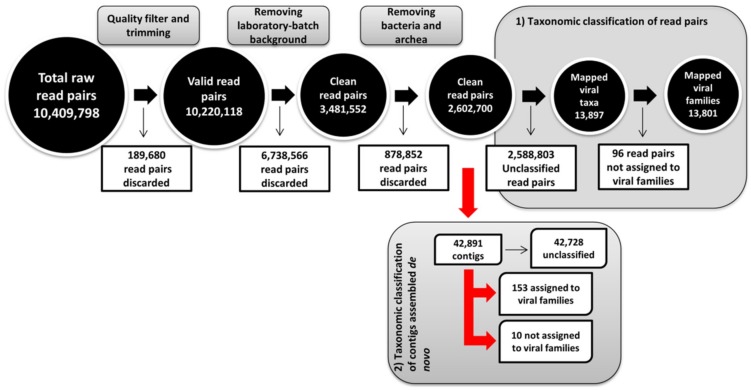
Next-generation sequencing (NGS) data analysis workflow. Metagenomic characterization workflow and taxonomic classification strategies of Illumina pair reads (1) and contigs assembled *de novo* (2). Pair reads quality filtering and trimming were performed with the bbduk program (BBTools v38.42). The centrifuge metagenomics classification system was used for the taxonomic classification of pair reads and contigs (Centrifuge version 1.0.3-beta) [43]. *De novo* assembly was performed using SPAdes v3.11 [45] and Unicycler.

**Figure 2 viruses-12-00422-f002:**
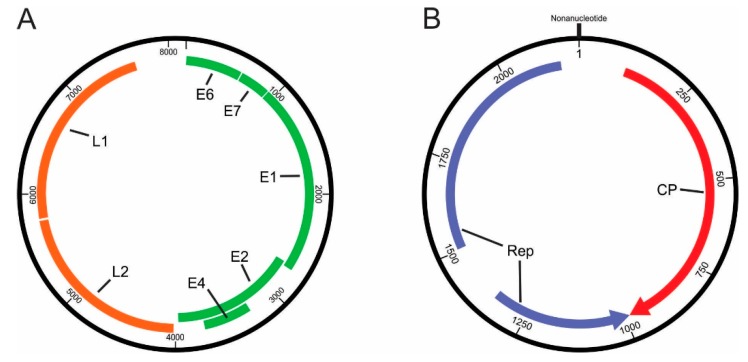
(**A**) *Tadarida brasiliensis* papillomavirus type 1 (TbraPV1) genome diagram indicating the positions of annotated PV genes. All genes are read from the direct strand. Genes encoding early and late proteins are depicted with green and orange lines, respectively. (**B**) *Tadarida brasiliensis* gemykibivirus 1 (TbGkyV1) genome diagram. Genes encoding the replication initiation protein (Rep) and the capsid protein (CP) are shown with blue and red arrows, respectively. Note that the *Rep* gene is on the reverse-complemented strand. The position of the nonanucleotide (TATAAATAG) is also indicated.

**Figure 3 viruses-12-00422-f003:**
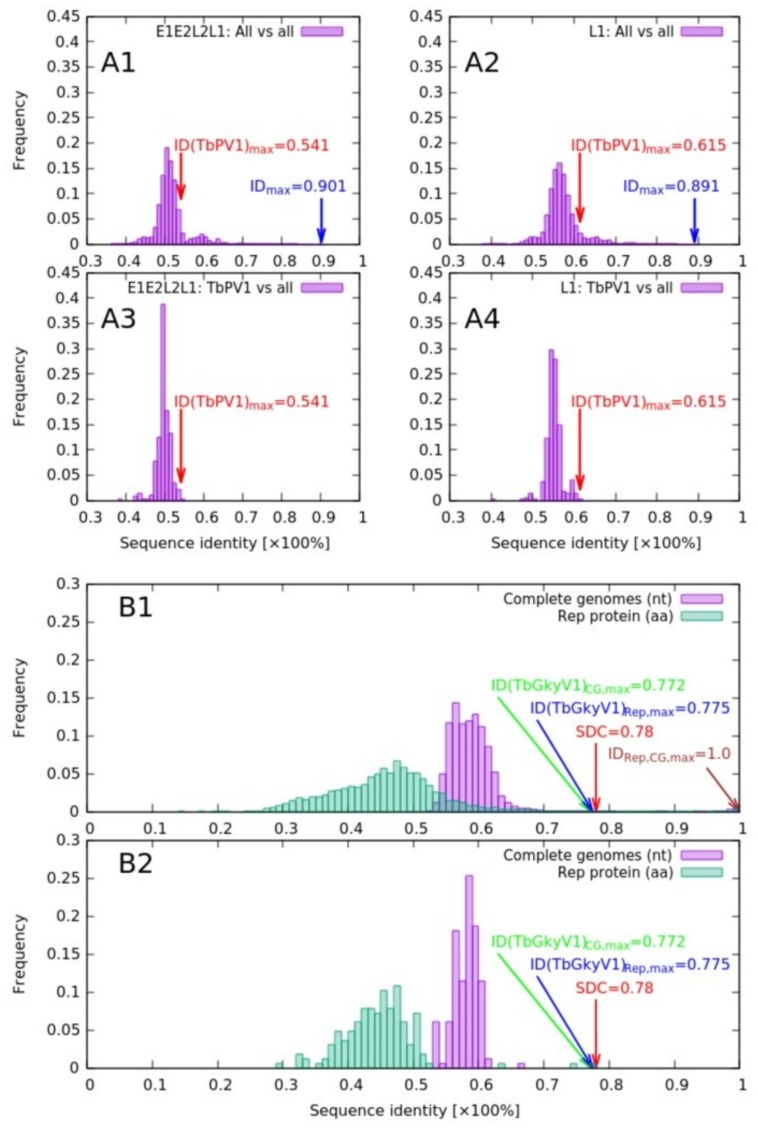
Frequency distribution of pairwise identity values from nucleotide sequence comparisons of PV and genomovirus genome regions. (**A**) Nucleic sequence identity histograms of the 377 PV nucleotide sequences (A1 and A2) and TbraPV1 (A3 and A4) based on the concatenated *E1, E2, L2, and L1* gene sequences (A1 and A3) and on the *L1* gene sequence (A2 and A4). Multiple sequence alignments were constructed using Muscle (v3.8.31) [47], and the distance matrices were estimated using SeaView v4.7 [49], as suggested in Bernard et al. (2010). Red arrows indicate the maximum sequence similarities of TbraPV1 for each of the sequence contexts (*E1, E2, L2, and L1* concatenation and the *L1* gene sequence). The blue arrows (A1 and A2) indicate the overall maximum sequence identity in the depicted context (*E1, E2, L2*, and *L1* concatenation and the *L1* gene sequence). The histograms were visualized using gnuplot (v5). (**B**) Pairwise sequence identity histograms based on 167 complete genome nucleotide (purple) and *Rep* gene (green) amino acid sequences from the *Genomoviridae* family, based on the entire pairwise identity distance matrices (B1) and on the matrix slices representing pairwise identities of TbGkyV1 to all other 166 *Genomoviridae* sequences (B2). The pairwise similarity matrices were obtained through pairwise sequence alignments (Muscle v3.8.31) [47], using Sequence Demarcation Toolkit (SDT v1) [55]. For the arrows in the histogram: red = the species demarcation threshold/criteria (SDC) for *Genomoviridae* [70]; green and blue = maximum pairwise identity values of TbGkyV1 for the complete genome (green) and the Rep protein sequence contexts (blue). Histograms were visualized using gnuplot (v5). nt = nucleotide, aa = amino acid, SCD = sequence demarcation threshold/criteria [70]. CG: complete genome.

**Figure 4 viruses-12-00422-f004:**
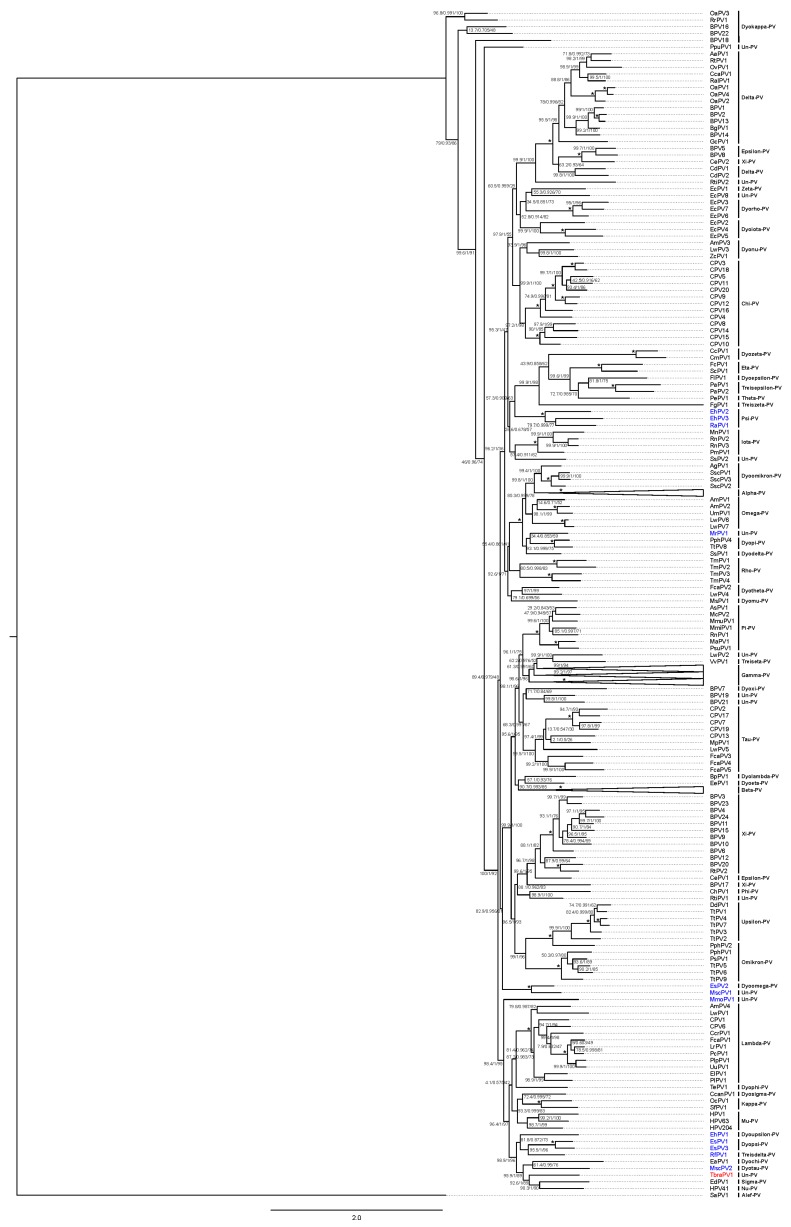
Phylogenetic tree of the PV *L1* gene. The tree was constructed using the GTR+F+R10 model, and branches are annotated with SH-aLRT (1000 replicates), aBayes, and UF bootstrap support (1000 replicates) values, respectively. Maximum support values are shown with asterisks (*). *Alpha*-, *Beta*-, and *Gamma*-PV genera were collapsed. Novel TbraPV1 is depicted in red. Bat PV types are depicted in blue. Un-PV: unclassified PV genera.

**Figure 5 viruses-12-00422-f005:**
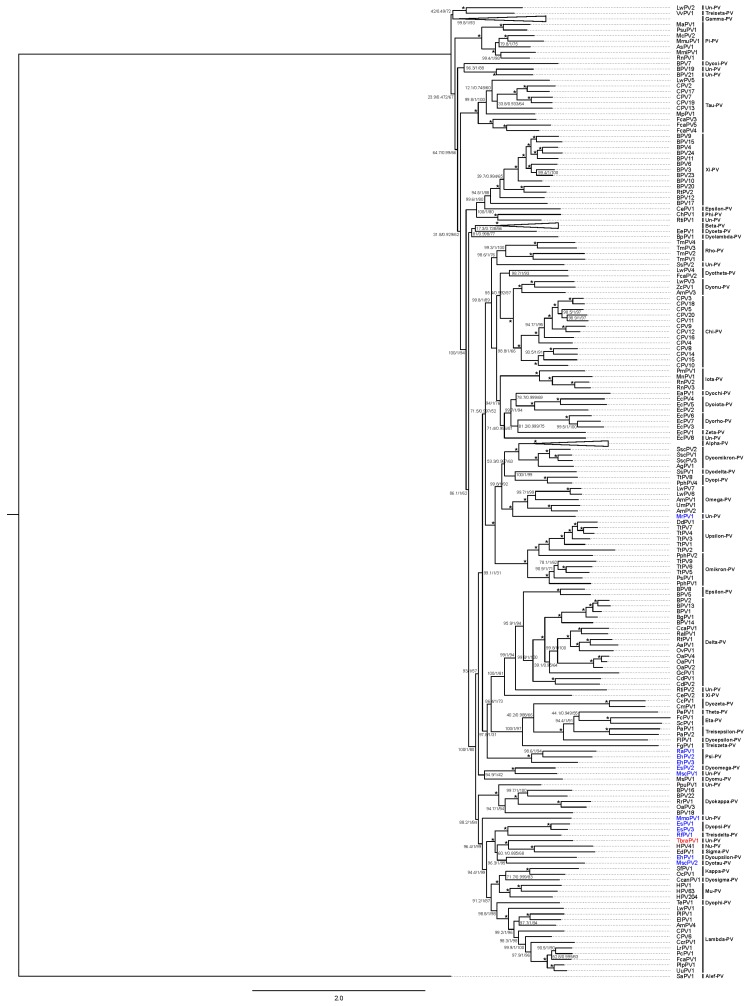
Phylogenetic tree of the concatenated *E1, E2, L2*, and *L1* PV genes. The tree was constructed using the GTR+F+G4 model, and branches are annotated with Sh-aLRT (1000 replicates), aBayes, and UF bootstrap support (1000 replicates) values, respectively. Maximum support values are shown with asterisks (*). *Alpha-*, *Beta-*, *and Gamma-PV* genera were collapsed. Novel TbraPV1 is depicted in red. Bat PV types are depicted in blue. Un-PV: unclassified PV genera.

**Figure 6 viruses-12-00422-f006:**
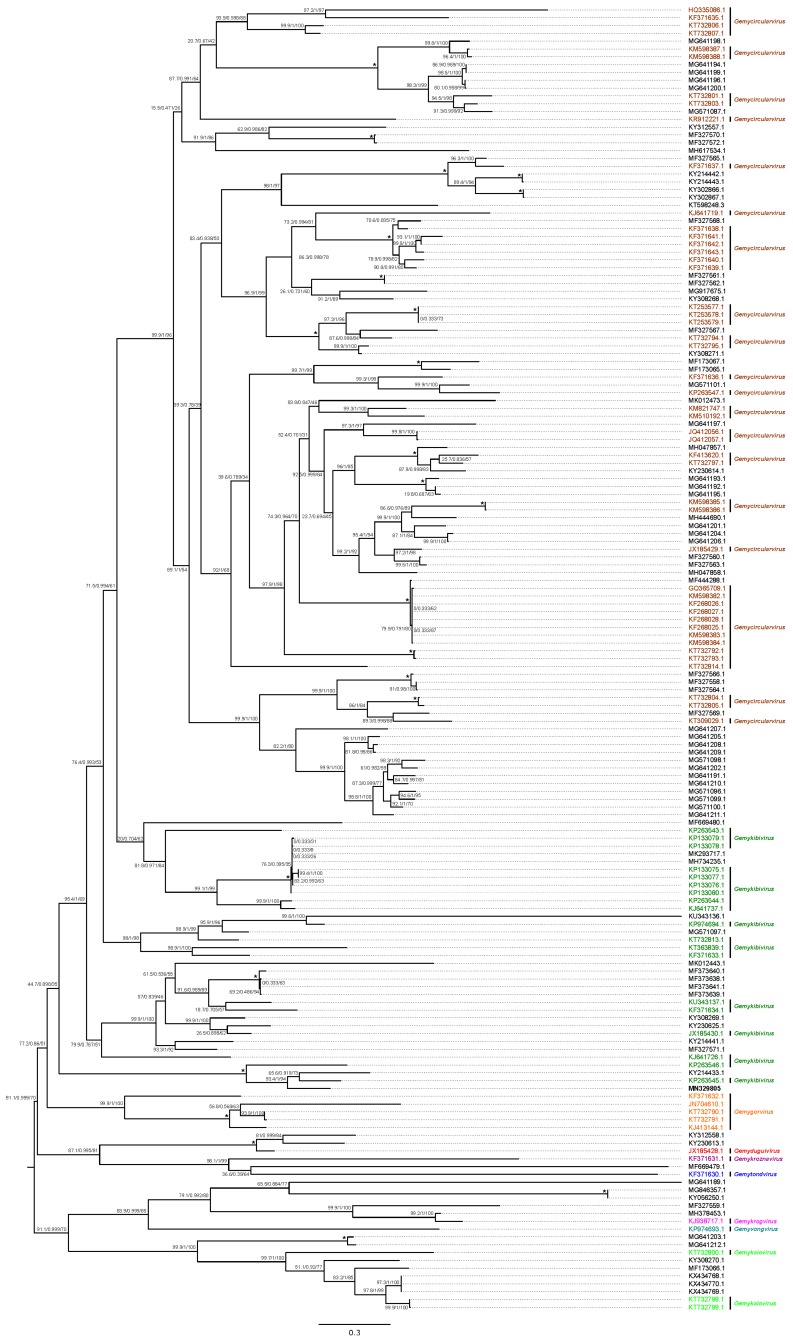
Phylogenetic tree of the Genomovirus Rep amino acid sequence. The tree was constructed using the LG+I+G4 substitution model, and branches are annotated with Sh-aLRT (1000 replicates), aBayes, and UF bootstrap support (1000 replicates) values, respectively. Maximum support values are shown with asterisks (*). Novel TbGkyV1 is depicted in bold. *Genomoviridae* genera were classified according to Varsani and Krupovic (2017). Unclassified sequences were not depicted.

**Table 1 viruses-12-00422-t001:** Read pairs and contigs of virus families identified in anal and oral swab samples of *Tadarida brasiliensis* obtained by metagenomics using Illumina technology.

Host	Family	No. of Read Pairs	No. of Contigs	Genome Type
Bacteria, Archea	*Inoviridae*	9908	8	ssDNA-C
	*Siphoviridae*	666	29	dsDNA-C
	*Myoviridae*	120	18	dsDNA-L
	*Podoviridae*	33	6	dsDNA-L
	*Rudiviridae*	6	0	dsDNA-L
	*Sphaerolipoviridae*	3	0	dsDNA-L
	*Microviridae*	1	0	ssDNA-C
	*Leviviridae*	1	0	ssRNA-L
Total	8	10,738	61	7 DNA; 1 RNA
Insects and other invertebrates	*Baculoviridae*	34	7	dsDNA-C
	*Ascoviridae*	34	0	dsDNA-C
	*Iridoviridae*	29	6	dsDNA-C
	*Nudiviridae*	15	0	dsDNA-C
	*Hytrosaviridae*	4	2	ssDNA-C
	*Dicistroviridae*	3	0	ssRNA-L
	*Polydnaviridae*	2	0	dsDNA-C
	*Solinviviridae*	2	3	ssRNA-L
	*Asfarviridae*	1	1	dsDNA-L
	*Nimaviridae*	1	0	dsDNA-C
Total	10	125	19	8 DNA; 2 RNA
Plants	*Phycodnaviridae*	42	6	dsDNA-C
	*Potyviridae*	10	0	ssRNA-L
	*Geminiviridae*	8	0	ssDNA-C
	*Caulimoviridae*	1	0	dsDNA-RT-C
	*Tymoviridae*	1	1	ssRNA-L
Total	5	62	7	3 DNA; 2 RNA
Protists	*Mimiviridae*	1825	0	dsDNA-L
	*Marseilleviridae*	65	1	dsDNA-C
	*Narnaviridae*	1	0	ssRNA-L
	*Totiviridae*	1	0	dsRNA-L
	*Pithoviridae*	1	0	dsDNA-C
Total	5	1893	1	3 DNA; 2 RNA
Vertebrates	*Retroviridae*	461	22	ssRNA-RT-L
	*Genomoviridae*	258	6	ssDNA-C
	*Herpesviridae*	113	4	dsDNA-L
	*Papillomaviridae*	90	14	dsDNA-C
	*Poxviridae*	36	6	dsDNA-L
	*Alloherpesviridae*	10	1	dsDNA-L
	*Rhabdoviridae*	3	2	ssRNA-L
	*Reoviridae*	3	0	dsRNA-L
	*Astroviridae*	2	0	ssRNA-L
	*Circoviridae*	2	0	ssDNA-C
	*Adenoviridae*	1	0	dsDNA-L
	*Polyomaviridae*	1	0	dsDNA-C
	*Coronaviridae*	1	0	ssRNA-L
	*Paramyxoviridae*	1	0	ssRNA-L
	*Flaviviridae*	1	0	ssRNA-L
Total	15	983	55	8 DNA; 7 RNA
Total sequences assigned to viral families	43	13,801	143	29 DNA; 14 RNA
Viral sequences not assigned to families		96	10	
Total viral sequences		13,897	153	

ss = single strand, ds = double strand, C = circular, L = linear.

**Table 2 viruses-12-00422-t002:** Main genomic features and putative proteins of the novel *Tadarida brasiliensis* papillomavirus type 1.

Genomic Regions/ORFs	Length (nt)	Nucleotide Sequence (Pre-Stop Codon) (nt)	Protein Size (aa)	HPV Motifs and Domains (Consensus Sequences)	Nucleotide Position	Amino Acid Position
*URR*	416	7735–8151		Polyadenylation site [AATAAA]	7700–7705	
					7906–7911	
					7924–7929	
				TATA box [TATAAA]	7686–7691	
					7824–7829	
				E1-binding site [CCATGAGAAATTGTTGTT]	8038–8055	
				E2-binding site [ACC(N)6GGT]	7885–7896	
					7934–7945	
					7992–8003	
*E6*	603	1–603	200	Zinc-binding domain [CXXC(X)_29_CXXC]	256–366	86–122
					475–585	159–195
				PDZ-binding domain [X(T/S)X(L/V)]	136–147	46–49
					202–213	68–71
					238–249	80–83
					277–228	93–96
*E7*	327	605–931	108	Zinc-binding domain [CXXC(X)_29_CXXC]	794–907	64–101
				pRB-binding site [(LXCXE)]	668–682	22–26
*E1*	1848	934–2781	615	Bipartite-like NLS [KRK(X)_24_KXXK]	1180–1272	83–113
				NES	1213–1242	94–103
				[L(X)_2-3_L(X)_2_(L,I,V)X(L,I)]		
				ATP-binding site [GXXXXGK(T/S)]	2260–2283	443–450
				Cdk-phosphorylation site [(S/T)P]	1216–1221	95–96
					1825–1830	298–299
					2362–2367	477–478
					2623–2628	564–565
					1198–1203	89–90
					1528–1533	199–200
					1243–1248	104–105
*E2*	1248	2717–3964	415	Leucine zipper domain [L(X)_6_L(X)_6_L(X)_6_L]	Absent	Absent
*E4*	426	3300–3725	141	Leucine motif [LLXLL]	Absent	Absent
*L2*	1758	4019–5776	585	Polyadenylation site [AATAAA]	5232–5237	
				Furin cleavage motif [RX(K/R)R]	5744–5758	576–579
				Transmembrane-like domain [G(X)_26_G]	4187–4270	57–84
*L1*	1548	5787–7334	516			

**Table 3 viruses-12-00422-t003:** Main genomic features and putative proteins of the novel *Tadarida brasiliensis* gemykibivirus 1.

Genome Regions	Length (nt)	Nucleotide Sequence (nt)	Protein Size (aa)	Motifs and Domains (Consensus Sequences)	Nucleotide Position	Amino Acid Position
**Large intergenic region**	69	2162–2196,1–36		Nonanucleotide at the putative stem-loop	1–9	
WATAWWHAN
Replicase (Rep)	Catalytic domain	668	2162–1494	223	Motif I [uuTYxQ]	2073–2090	25–30
Motif II [xHxHx]	1965–1979	62–66
Motif III [YAxK]	1842–1853	104–107
GRS domain [(x)_2_FD(x)_4_HPN(x)_5_]	1875–1925	80–95
Central domain	375	1381–1006	125	Walker A [G(x)_4_GKT]	1325–1348	7–14
Walker B [xFDDx]	1220–1234	45–49
Walker C [W/Y(x)_2_N]	1091–1102	89–92
Capsid (Cap/CP)	974	36–1010	324

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
