# Peer review of "A Preliminary Study of the Virome of the South American Free-Tailed Bats (Tadarida brasiliensis) and Identification of Two Novel Mammalian Viruses"

_viruses, 2020, doi:10.3390/v12040422_

Round 1
Reviewer 1 Report
Summary of the manuscript:
Study implemented sequence-independent high throughput sequencing approach to identify the viral composition of 1 pool of rectal/oral swabs collected from female members of a T. brasiliensis colony. The pool contained 5 swabs that were positive initially with a PCR for papillomaviruses (total 49 rectal and 49 oral swabs tested). Using additional conventional PCR means, the complete genome of the papillomavirus obtained with bioinformatics was confirmed. Additionally the genome (approximately 2000bp) of a Genomovirus was also recovered. The complete genomes of both viruses are described in detail, highlighting gene and regulatory features. Papillomaviruses are highly diverse viruses, with additional metagenomic studies, it seems more likely that the known diversity of these viruses will continue to expand. Thus, detailed description of the identified genome is important for comparison studies and formal taxonomic classification.
Appropriately, the authors highlight the need for expanding the known diversity of viral agents harbored by New World bat species. It is also appreciated that the authors make a strong point to highlight the importance of bats in ecosystem services and that little research has investigated their own pathogens. Likewise, a good description for the need to add various controls during high throughput sequencing applications.
The concern I have regarding the manuscript is in its reporting/ description of the “virome” (line 381). The viral composition is not sufficiently described. The materials and methods don’t adequately describe what the authors did. For example, how the manuscript reads was that the swabs were briefly centrifuged and the pellet treated with proteinase K (swabs were collected and placed in the saline solution, this was then centrifuged briefly, resuspended and treated with proteinase K), RCA was applied and sequenced with Illumina. No extraction procedure is described, can the authors please provide a reference for this approach? Also the speed of centrifugation (13,000 xg for 5 minutes) seems to be a speed to use to pellet large debris and cells. According to all approaches of metagenomic studies, this would likely have decreased the yield of viral particles in the sample material that they used for RCA. Perhaps the authors can provide a reference to back up their choice? Multiple reviews and investigations report that RCA does introduce bias to DNA viruses, though the authors did acknowledge this.
The other major issue, which has been highlighted in a recent bat viral metagenomic review, is the reporting of viral families detected in metagenomic studies with no actual descriptions of the viral sequences detected. This is in specific reference to the majority of mammalian-infecting virus families reported in the manuscript. Some of these families are even reported to be detected in South American bats for the first time (line 446-447), however, these detections are not described (rhabdoviruses, coronaviruses, poxviruses, flaviviruses, herpesviruses and paramyxoviruses to name only a few). The authors even acknowledge the value in caution when comparing identified reads to (for example) uncurated genbank entries. Without a description of what was detected, and perhaps strengthened with followed up with conventional PCR methods, how can the data be evaluated and supported? There is a table listing what was detected, but this is in no way sufficient to support the findings. If these other sequences are deposited anywhere needs to be indicated; likewise, if follow up investigations will be published reporting detailed descriptions of these viruses (but then the focus of the manuscript should be changed). The effect of omitting these findings would be the same as if they are kept in the manuscript as they are currently reported - no sequence comparisons can be made with the data, no fundamental comparison between diversity of viruses in this species can be made to others.
Alternatively, the authors should move away from the “metagenomic/virome” aspect of the study and focus on what seems to have been the priority in this study – papillomaviruses (and the identified Genomovirus as a nice coincidence). The methods used seems to be biased towards these viruses (few samples, pre-screening, DNA viruses, rolling circle replication etc). The detailed analysis of these viruses add value for expansion of the evolution of the Papillomaviridae family (a highly diverse family of viruses thought to be rather host-specific) and individuals in this field or others identifying such viruses with unbiased approaches will undoubtedly find this manuscript useful. I would suggest better descriptions of the papillomavirus taxonomic cut offs for species, genus, subfamily etc. in the manuscript (lines 463-464: summarize rather) as done with the Genomoviridae lines 480-481).
Furthermore, has the raw high throughput sequencing data been deposited in a public database?
Specific suggestions per line numbers:
46-47: the paragraph separation should be corrected, the first paragraph is just one sentence long.
52-53 : “such as severe acute respiratory syndrome (SARS)-like coronavirus (SL-CoV),”; suggest changing to “severe acute respiratory syndrome (SARS)-related coronaviruses“
55-58: “In addition, the gregarious behavior of many bat species, such as free-tailed bats Tadarida brasiliensis (I. Geoffroy Saint-Hilaire, 1824)—which are the most abundant migratory and cosmopolitan species of the New World bats, and are widespread throughout the Americas [11–13] and protected by international agreements [14]—may facilitate rapid transmission of pathogens between bats and other species [6].” ; this sentence is very long and a little difficult to follow; please clarify or restructure.
60: is the use of (geno)types necessary? Why not simply genotypes?
60-61: reference style does not conform to the rest of the manuscript.
84: no morphological confirmation with cytochrome barcoding?
85: please provide information on the swabs used- such as manufacturer? Size? Etc
95-97: please provide a supporting reference for this approach? Or perhaps reasoning for this methodology?
111-114 and 200 and 398: relating to the laboratory background removal with the kmers – please clarify the reasoning and methodology for this. Lines 111-114, is very unclear of how kmers/indexes are used. Are the kmers added during the RCA? Are the kmers indexes? Are the kmers arbitrarily selected sequences (as in kmers used in Victoria et al. 2009)? Or fixed 27 nt sequences? The purpose/explanation of why this was done seems clear, but not how it was done. Removal of 6.7 mil reads also seems like a lot, thus a better explanation from the methods through to the discussion would make this more understandable (line 398 explains this better, but it needs to come across in the methods).
638: reference 41 – is missing the journal information
Author Response
Summary of the manuscript:
Study implemented sequence-independent high throughput sequencing approach to identify the viral composition of 1 pool of rectal/oral swabs collected from female members of a T. brasiliensis colony. The pool contained 5 swabs that were positive initially with a PCR for papillomaviruses (total 49 rectal and 49 oral swabs tested). Using additional conventional PCR means, the complete genome of the papillomavirus obtained with bioinformatics was confirmed. Additionally the genome (approximately 2000bp) of a Genomovirus was also recovered. The complete genomes of both viruses are described in detail, highlighting gene and regulatory features. Papillomaviruses are highly diverse viruses, with additional metagenomic studies, it seems more likely that the known diversity of these viruses will continue to expand. Thus, detailed description of the identified genome is important for comparison studies and formal taxonomic classification.
Appropriately, the authors highlight the need for expanding the known diversity of viral agents harbored by New World bat species. It is also appreciated that the authors make a strong point to highlight the importance of bats in ecosystem services and that little research has investigated their own pathogens. Likewise, a good description for the need to add various controls during high throughput sequencing applications.
The concern I have regarding the manuscript is in its reporting/ description of the “virome” (line 381). The viral composition is not sufficiently described. The materials and methods don’t adequately describe what the authors did. For example, how the manuscript reads was that the swabs were briefly centrifuged and the pellet treated with proteinase K (swabs were collected and placed in the saline solution, this was then centrifuged briefly, resuspended and treated with proteinase K), RCA was applied and sequenced with Illumina. No extraction procedure is described, can the authors please provide a reference for this approach? Also the speed of centrifugation (13,000 xg for 5 minutes) seems to be a speed to use to pellet large debris and cells. According to all approaches of metagenomic studies, this would likely have decreased the yield of viral particles in the sample material that they used for RCA. Perhaps the authors can provide a reference to back up their choice? Multiple reviews and investigations report that RCA does introduce bias to DNA viruses, though the authors did acknowledge this.
Our research group has extensive experience with research of papillomavirus (PV) infections, both in humans (Chouhy et al., 2010; Chouhy et al., 2013; Bolatti et al., 2017; Bolatti & Hosnjak et al., 2018) and animals (Kocjan et al., 2017), in several distinct sample types, including cutaneous/mucosal swabs. The primary purpose of this study was focused on the identification of novel papillomaviruses infecting T. brasiliensis, in order to explore the diversity of PV in different hosts. Therefore, swab samples included in this work were first processed using experimental protocols, designed previously to suit our aforementioned initial aim (Chouhy et al., 2010; Chouhy et al., 2013; Bolatti et al., 2017; Bolatti & Hosnjak et al., 2018). In addition, viral DNA enrichment using RCA has been widely used by other researchers in studies aimed to explore the virome composition of different types of samples, originating from bats and other animals (Yinda et al., 2016; Van Doorslaer et al., 2017; Varsani et al., 2014). On the other hand, and as the reviewer pointed out, we recognize the weaker points of our approach and we have addressed them in the Discussion section (lines 415-424 of the original manuscript and lines 439-452 of the revised manuscript), in order for them to be improved in (our) future studies.
We agree with Reviewer 1 that information regarding the sample processing protocol and the corresponding references should be provided. Accordingly we have included this information in the revised manuscript, as follows (Lines 105-114):
Samples were processed according to previously published protocols that have been successfully applied for identification of papillomavirus (PV) in human skin swab samples [35–37]. Briefly, the cells were centrifuged at 13,000 × g for 5 min and the pellets were resuspended in 100 μl TE buffer (Qiagen, Hilden, Germany), containing 100 μg of proteinase K (Qiagen), and incubated overnight at 55 °C. Following proteinase K inactivation (95 °C for 10 min), the lysates were stored at −20 °C. Subsequently, the obtained samples were tested for the presence of PV DNA, using improved versions of FAP [38,39] and CUT PCRs [36], as described previously [37,40]. Circular DNA molecules in lysates of five selected PV-positive samples (four anal and one oral swab) were enriched using rolling-circle amplification (RCA) using the illustra TempliPhi 100 Amplification Kit (GE Healthcare, Chicago, IL, USA) [41–43].
The other major issue, which has been highlighted in a recent bat viral metagenomic review, is the reporting of viral families detected in metagenomic studies with no actual descriptions of the viral sequences detected. This is in specific reference to the majority of mammalian-infecting virus families reported in the manuscript. Some of these families are even reported to be detected in South American bats for the first time (line 446-447), however, these detections are not described (rhabdoviruses, coronaviruses, poxviruses, flaviviruses, herpesviruses and paramyxoviruses to name only a few). The authors even acknowledge the value in caution when comparing identified reads to (for example) uncurated genbank entries. Without a description of what was detected, and perhaps strengthened with followed up with conventional PCR methods, how can the data be evaluated and supported? There is a table listing what was detected, but this is in no way sufficient to support the findings. If these other sequences are deposited anywhere needs to be indicated; likewise, if follow up investigations will be published reporting detailed descriptions of these viruses (but then the focus of the manuscript should be changed). The effect of omitting these findings would be the same as if they are kept in the manuscript as they are currently reported - no sequence comparisons can be made with the data, no fundamental comparison between diversity of viruses in this species can be made to others.
We appreciate the observation of Reviewer 1 and agree that a more detailed analysis of viral nucleotide sequences in bat metagenomic studies should be of greater interest in the future and that we should aim at a quality of results, which are comparable at least to those in modern bacterial metagnomics studies, operating at the level of metagenome assembled genomes (MAGs). However, it should be pointed out that due to small genome sizes of viruses, even the slightest addition of bacterial “contamination” to a sample could hinder the detection of viruses. This reflects the relatively very low counts of viral reads in our (and in nearly all similar studies), which for the most part hinders the construction of viral MAGs – which is also likely the case in our study: we only obtained 153 viral MAGs, which were widely distributed among different viral families.
In this study, also due to the overall low yield of viral reads and contigs, we decided to limit the detailed analyses to viral sequences that represented complete viral genomes. Reviewer 1 also specifically mentions certain vertebrate viral families, where he/she finds hindrances in our article. Regarding those, we would like to point out that while we did assemble some sequences similar to Pox, Herpes and Alloherpesviridae, they were only represented by short sequence fragments that may have been misclassified or may simply be part of the host genome, similar to those families of viruses (the genome of T. brasiliensis has not been sequenced yet). On the other hand the vertebrate RNA viruses (mentioned by Reviewer 1), were only represented by extremely low read-pair counts, which also may have been misclassified. In view of these specific questions posed by Reviewer 1, the following paragraph has been added to the Discussion section of the revised manuscript (lines 532-555):
Finally, it is worth noting that the results of our metagenomic screening of the pooled T. brasiliensis oral and anal swab samples is effectively provided at three different levels of specificity/sensitivity. The two complete genome sequences (TbraPV1 and TbGkyV1), that have been described at the highest level of detail, also confer the highest level of confidence. In other words, we have complete confidence that these two viruses were present in the pooled nucleic acid samples. On the other hand, we would expect that many viruses that were not assembled to the level of complete genomes were therefore not represented in this group of results. Assigning sequences that did not assemble at the level of complete viral genomes to taxonomical families could therefore be a valid approach offering a higher level of sensitivity than only the complete genome assemblies, but at the cost of diminished specificity. By definition, a great deal of specificity was already lost by summarizing the taxonomical mappings upwards to the family level. In addition, identifying a sequence fragment that resembles a known viral genome in a given genomic region, may not always be sufficient to infer that that specific virus was present in the sample. Viruses are highly promiscuous entities, which can easily exchange parts of their genomes with their hosts, be integrated or even naturalized into the host genomes [107]. Taxonomical viral families identified among the assembled contigs could be interpreted as viral families that were probably represented in our samples. Lastly, and due to the low number of assembled contigs, that were found related to known viral sequences, we attempted to increase the sensitivity even further by obtaining taxonomical family mappings also for the source read pairs. These results, however, should be interpreted with utmost caution, because they likely confer a very low level of specificity due to the limited sequence length (2 × 150 bp). taxonomical viral families identified by read-pair taxonomy mapping only, merely suggest the possibility that these families were present and should be replicated in the future by taxonomical mappings conferring greater specificity, for example with longer sequences, such as those assembled from illumina or nanopore reads.
In addition, to facilitate future comparisons of our results with newly obtained information, the assembled sequences that were used in the taxonomy mapping were added as a supplementary data to the revised manuscript (lines 210-212), as follows:
The contigs, obtained by de novo assembly as part of the metagenomic workflow (2) have also been made available for download (Supplementary Data 1).
In addition, the corresponding heading was included at the Supplementary Material section of the revised manuscript (lines 570-572), as follows:
Supplementary Data 1. Details of the assembled nucleotide sequences obtained by de novo assembly and used in the taxonomic classification as a part of the metagenomic workflow (2) (Figure 1).
Alternatively, the authors should move away from the “metagenomic/virome” aspect of the study and focus on what seems to have been the priority in this study – papillomaviruses (and the identified Genomovirus as a nice coincidence). The methods used seems to be biased towards these viruses (few samples, pre-screening, DNA viruses, rolling circle replication etc). The detailed analysis of these viruses add value for expansion of the evolution of the Papillomaviridae family (a highly diverse family of viruses thought to be rather host-specific) and individuals in this field or others identifying such viruses with unbiased approaches will undoubtedly find this manuscript useful. I would suggest better descriptions of the papillomavirus taxonomic cut offs for species, genus, subfamily etc. in the manuscript (lines 463-464: summarize rather) as done with the Genomoviridae lines 480-481).
We appreciate the suggestion of Reviewer 1, however, we believe that the description of viral sequences present in T. brasiliensis, using our approach, represents an original and valuable contribution to the scientific community and is in fact also the first preliminary study of viruses circulating in this species. We agree that our study could represent an overview of the T. brasiliensis virome and, as suggested by Reviewer 2, we have modified the title and some aspects of the revised manuscript to better clarify such statements, as follows:
Title (lines 2-5): “A preliminary study of the virome of the South American free-tailed bats (Tadarida brasiliensis) and identification of two novel mammalian viruses”
Introduction (lines 72-75): In this study we report a detailed description of two novel complete genome sequences, one describing a new papillomavirus genus and the other representing a novel variant of an existing gemykibivirus species. In addition, we report a preliminary overview of the T. brasiliensis virome composition.
Discussion (lines 399-400): Here, a first attempt to assess the virome composition in pooled oral and anal swabs of T. brasiliensis was presented.
Conclusion (lines 558-560): This study presents an initial description of the oral/anal virome composition of T. brasiliensis, a widely-distributed New World bat species living in close contact with the human population, for the first time.
We also agree that more details of Papillomaviridae taxonomic classification should be provided. Accordingly, this information has been introduced to the discussion section of the revised manuscript, as follows (lines 491-502):
According to the current ICTV Papillomaviridae classification guidelines (published in June 2018), based on the nucleotide sequence of the L1 gene [97], TbraPV1 is the founding member of a novel PV genus in the Firstpapillomavirinae taxonomical subfamily, sharing more than 45% sequence identity to other PV types included in this subfamily. Although nucleotide sequence analysis in the L1 gene indicated that TbraPV1 shares a 61.5% identity with HPV41 (Nu-PV) and should be included within the same genus (more than 60% of nucleotide identity in L1 gene) [92], the mentioned demarcation criteria suggests a visual inspection of phylogenetic trees derived from concatenated E1, E2, L1, and L2 nucleotide sequences to delineate PV genera [97]. In the present study, such analysis clustered TbraPV1 basal to the delineation of HPV41 (Nu-PV) and EdPV1 (Sigma-PV), identified in a North American porcupine (Erethizon dorsatum) and, therefore, may represent a novel genus within the Papillomaviridae family.
Furthermore, has the raw high throughput sequencing data been deposited in a public database?
The high throughput sequencing data obtained in this study, that was used for the metagenomic reconstruction, has been deposited at the NCBI Sequence Read Archives (SRA) (https://www.ncbi.nlm.nih.gov/sra/) and the corresponding accession number (PRJNA615356) has been included in the revised version of the manuscript, as follows (lines 208-212):
The relevant raw high throughput sequencing data obtained in this study was deposited at the NCBI Sequence Read Archives (SRA) with the following accession number: PRJNA615356. The contigs, obtained by de novo assembly as part of the metagenomic workflow (2), have also been made available for download (Supplementary Data 1).
Specific suggestions per line numbers:
46-47: the paragraph separation should be corrected, the first paragraph is just one sentence long.
As we agree with the suggestion of Reviewer 1, the mentioned modification was included in the revised version of the manuscript, as follows (lines 48-52):
Bats belong to the order Chiroptera, which is the second-largest mammalian group, comprising 21 families and 1,411 species distributed globally, with the exception of polar areas [1,2]. Approximately 25% of the world’s bat species are endangered, causing concerns about the negative conservation impact and its influence on the ecosystem services these bats provide, such as arthropod regulation, seed dispersal, and pollination [1,3].
52-53: “such as severe acute respiratory syndrome (SARS)-like coronavirus (SL-CoV),”; suggest changing to “severe acute respiratory syndrome (SARS)-related coronaviruses“
We agree with the Reviewer 1 and have therefore corrected the mentioned expression (line 56), as follows:
»...such as severe acute respiratory syndrome (SARS)-related coronavirus,...«
55-58: “In addition, the gregarious behavior of many bat species, such as free-tailed bats Tadarida brasiliensis (I. Geoffroy Saint-Hilaire, 1824)—which are the most abundant migratory and cosmopolitan species of the New World bats, and are widespread throughout the Americas [11–13] and protected by international agreements [14]—may facilitate rapid transmission of pathogens between bats and other species [6].” ; this sentence is very long and a little difficult to follow; please clarify or restructure.
According to the suggestion of Reviewer 1, we have corrected the questionable sentence in the revised version of the manuscript, as follows (lines 57-62):
In addition, the gregarious behavior of many bat species, such as free-tailed bats Tadarida brasiliensis (I. Geoffroy Saint-Hilaire, 1824), may facilitate rapid transmission of pathogens between bats and other species [6]. T. brasiliensis is the most abundant migratory and cosmopolitan species of the New World bats, widespread throughout the Americas [11–13] and protected by international agreements [14].
60: is the use of (geno)types necessary? Why not simply genotypes?
According to the suggestion of Reviewer 1 we have changed the questionable term to “genotypes” in the revised version of the manuscript (line 64).
60-61: reference style does not conform to the rest of the manuscript.
We would like to thank Reviewer 1 for his observation. References style for these citations [17,18] have been corrected in the revised version of the manuscript (line 65).
84: no morphological confirmation with cytochrome barcoding?
We appreciate the Reviewer´s suggestion. We believe that species confirmation of T. brasiliensis by means of cytochrome barcoding is not necessary, since the species is very conspicuous, being the only one in Argentina with exclusive and diagnostic characteristics, preventing misclassification for any other species of bats. In particular, T. brasiliensis can be identified by their medium size, dark grayish brown skin coloration and skull characteristics with a dental formula that includes convergent upper incisors. Another indicative characteristic of T. brasiliensis in Argentina are their deeply wrinkled lips which are not present in any other similar species of bats in the country. Furthermore, to our knowledge, there are currently no bibliographical references for cytochrome barcoding of T. brasiliensis from Argentina.
85: please provide information on the swabs used- such as manufacturer? Size? Etc
According to the suggestion of Reviewer 1, we provided details regarding the swabs used for our sampling process, as follows (lines 94-96 of the revised manuscript):
The oral cavity and anal regions of each individual were sampled using individual sterile cotton-tipped swabs (Deltalab, Barcelona, Spain), rolled back and forth (10 times), suspended in 200 µl of saline solution (NaCl 0.9%), and stored at 4 °C until further processing.
95-97: please provide a supporting reference for this approach? Or perhaps reasoning for this methodology?
As explained in the first comment to Reviewer 1, initially, this study was indeed focused on the identification of novel PV infecting T. brasiliensis, in order to explore their diversity in different hosts. Therefore, swab samples included in this work were first processed using experimental protocols, designed previously to suit our aforementioned initial aim (Chouhy et al., 2010; Chouhy et al., 2013; Bolatti et al., 2017; Bolatti & Hosnjak et al., 2018) and the viral DNA was enriched using RCA, previously also applied by others (Yinda et al., 2016; Van Doorslaer et al., 2017; Varsani et al., 2014).
As mentioned above, we agree that a more detailed information regarding our sample processing protocol and the corresponding references should be provided in the revised manuscript. Thus, we have provided the following modifications (Lines 105-114):
Samples were processed according to previously published protocols that have been successfully applied for the identification of papillomavirus (PV) in human skin swab samples [34–36]. Briefly, the cells were centrifuged at 13,000 × g for 5 min and the pellets were resuspended in 100 μl TE buffer (Qiagen, Hilden, Germany), containing 100 μg of proteinase K (Qiagen), and incubated overnight at 55 °C. Following proteinase K inactivation (95 °C for 10 min), the lysates were stored at −20 °C. Subsequently, the obtained samples were tested for the presence of PV DNA using improved versions of FAP [37,38] and CUT PCRs [35], as described previously [36,39]. Circular DNA molecules in lysates of five selected PV-positive samples (four anal and one oral swab) were enriched using rolling-circle amplification (RCA) using the illustra TempliPhi 100 Amplification Kit (GE Healthcare, Chicago, IL, USA) [40–42].
To clarify our experimental approach, an additional paragraph was included in the Discussion section of the revised manuscript (445-452):
Initially, this study was focused on the identification of novel PVs in T. brasiliensis, in order to explore their diversity in different hosts. Accordingly, the swab samples included in this work were first processed using experimental protocols, designed previously to suit our aforementioned initial aim [34–36]. Viral DNA was enriched using RCA, as suggested previously by others [40-42]. However, it should be noted that RCA may have favorably facilitated the amplification of circular genomes and, as a consequence, hindered the detection of linear genomes. Thus, more than 80% of the classified viral sequences (11,162/13,801) identified in our analysis corresponded to circular viral genomes.
111-114 and 200 and 398: relating to the laboratory background removal with the kmers – please clarify the reasoning and methodology for this. Lines 111-114, is very unclear of how kmers/indexes are used. Are the kmers added during the RCA? Are the kmers indexes? Are the kmers arbitrarily selected sequences (as in kmers used in Victoria et al. 2009)? Or fixed 27 nt sequences? The purpose/explanation of why this was done seems clear, but not how it was done. Removal of 6.7 mil reads also seems like a lot, thus a better explanation from the methods through to the discussion would make this more understandable (line 398 explains this better, but it needs to come across in the methods).
We would like to thank Reviewer 1 for pointing out the lack of clarity of the manuscript on this point. It is very important to us that this part of the manuscript is understood by the reader. What we refer to as “k-mers” are simply subsequences or substrings of the sequencing reads of length k=27. They are obtained by sliding a window of length k through each sequencing read and saving the content of that window. Then the datasets are compared to each other in respect of this kmer content. Reads in the metagenomic dataset that contained kmers (in other words, traces) that also appear in the co-processed datasets (the ones not related to this study, but that were processed alongside in the same sequencing batch) were removed.
For greater clarity, k-mers of length k=3 in the word “America” are: “Ame”, “mer”, “eri”, “ric” and “ica”. The concept of k-mers is widely used in bioinformatics, in applications such as short-read de novo assembly in which they represent the elementary operational unit, on which the de Brujin-graph construction is based. In sequence filtering vs. another sequence database, the use of k-mers can reduce sensitivity to sequencing error. If in the example above, due to a sequencing error the word “America” would have been read as “Imerica” only one out of five 3-mers would have been changed and the word would still be removed from the “metagenomic” dataset.
Please note that the following paragraphs have been modified in the M&M section, lines 124-130, as follows:
The read pairs contained in the metagenomic sample, which shared k-mers, sliding-window subsequences of 27 nt, with the sequencing datasets of samples (six in total) that were processed and analyzed in the same sequencing batch, were discarded using the bbduk program (referred to as laboratory-batch background screen in Figure 1). The primary purpose of this step was to conservatively limit the possibility of falsely identifying viral taxa that did not originate from the bat metagenomics sample and that could have been introduced by aerosol during sample processing or index hopping during sequencing.
In order to draw the reader’s attention to this matter in the “Discussion” section we further modified the paragraph at lines 408-409 by prepending the sentence:
During our study a total of 6,738,566 read pairs were removed during the laboratory batch-background screening, due to traces of sequences from co-processed samples.
638: reference 41 – is missing the journal information
We would like to thank Reviewer 1 for his observation. Journal information for this reference [44] has been corrected in the revised version of the manuscript (line 713).
Reviewer 2 Report
This paper undertook a metagenomics approach to characterize the viral families in fecal and oral swab samples of Brazilian free-tailed bats (Tadarida brasiliensis). The data described herein represent novel data for South American bats, for which little is known about their virome. In addition to a tabular presentation of the viral families detected in the bats (read pairs or contigs), the authors report the complete genomes of two novel viruses – Tadarida brasiliensis papillomavirus type 1 [TbraPV1] and Tadarida brasiliensis gemykibivirus 1 [TbGkyV1] which have been submitted to GenBank. The paper is well written, and the molecular analysis is very thoroughly performed. Steps were taken to prevent against contamination and to ensure data quality; for genome assemblies, contigs were mapped back to the genome and Sanger sequencing was also performed as quality checks. Data are provided regarding the phylogenetic placement of the novel viruses, and details on the genomic organization and coding regions. The results are of interest to the readership of Viruses. My specific comments are below.
1. Section 2.2, lines 94 – 102. Can you clarify why the first step was to test the samples specifically for papillomavirus DNA? Was this to guide selection of a subset of samples for next generation sequencing in which some viral nucleic acid was definitely present? If so, why did you start with papillomaviruses? I understand the general necessity to enrich for viral nucleic acid in such a complex sample, but I’m sure there was reasoning behind how this was done.
2. Table 1 – is it possible to add which viral reads/contigs were detected in the fecal samples and which were detected in the oral samples, or were the oral and fecal samples pooled together?
3. Table 1 – It strikes me that out of over 10 million read pairs that came out of the analysis, some of the viral families are a single read pair or 1 contig. The authors were very careful to take steps to prevent contamination (and to identify potential contamination, if it happened), and had quality assurance measures in place, but it’s difficult for me to know how to interpret a single (or few) read pairs in such a soup of information. Contigs were >500bp, but how long were the reads, for these instances where there was only a single read? Are you confident those reads are in parts of the genomes that are diagnostic to those viral families? How good were the matches between the reads and the viral families?
4. Title – because these data represent only 5 bats from a single location, oral/fecal data seem to be combined, and many viral families are represented by only 1 to a few read pairs, I would suggest adding the word “Preliminary” or “Initial” to the beginning of the title.
5. For the phylogenetic analyses (Figs 4, 5) – can you provide supplementary information noting the accession numbers and key to abbreviations for the sequences used in these analyses?
Author Response
Comments and Suggestions for Authors
This paper undertook a metagenomics approach to characterize the viral families in fecal and oral swab samples of Brazilian free-tailed bats (Tadarida brasiliensis). The data described herein represent novel data for South American bats, for which little is known about their virome. In addition to a tabular presentation of the viral families detected in the bats (read pairs or contigs), the authors report the complete genomes of two novel viruses – Tadarida brasiliensis papillomavirus type 1 [TbraPV1] and Tadarida brasiliensis gemykibivirus 1 [TbGkyV1] which have been submitted to GenBank. The paper is well written, and the molecular analysis is very thoroughly performed. Steps were taken to prevent against contamination and to ensure data quality; for genome assemblies, contigs were mapped back to the genome and Sanger sequencing was also performed as quality checks. Data are provided regarding the phylogenetic placement of the novel viruses, and details on the genomic organization and coding regions. The results are of interest to the readership of Viruses. My specific comments are below.
- Section 2.2, lines 94 – 102. Can you clarify why the first step was to test the samples specifically for papillomavirus DNA? Was this to guide selection of a subset of samples for next generation sequencing in which some viral nucleic acid was definitely present? If so, why did you start with papillomaviruses? I understand the general necessity to enrich for viral nucleic acid in such a complex sample, but I’m sure there was reasoning behind how this was done.
Our research group has extensive experience with research of papillomavirus (PV) infections, both in humans (Chouhy et al., 2010; Chouhy et al., 2013; Bolatti et al., 2017; Bolatti & Hosnjak et al., 2018) and animals (Kocjan et al., 2017), in several distinct sample types, including cutaneous/mucosal swabs. Initially, this study was indeed focused on the identification of novel PVs infecting T. brasiliensis, in order to explore their diversity in different hosts. Therefore, swab samples included in this work were first processed using experimental protocols, designed previously to suit our aforementioned initial aim (Chouhy et al., 2010; Chouhy et al., 2013; Bolatti et al., 2017; Bolatti & Hosnjak et al., 2018). In addition, in some previous studies aimed to explore the virome composition of different samples, including those focused on the identification of novel PV types in bats or other animals, enrichment of viral DNA using rolling circle amplification (RCA) has also been applied (Yinda et al., 2016; Van Doorslaer et al., 2017; Varsani et al., 2014).
As we are aware that our enrichment strategy was biased in favor of viruses with circular DNA genomes and to clarify our approach, an additional paragraph was included in the Discussion section of the revised manuscript (445-452):
Initially, this study was focused on the identification of novel PVs in T. brasiliensis, in order to explore their diversity in different hosts. Accordingly, swab samples included in this work were first processed using experimental protocols, designed previously to suit our aforementioned initial aim [34–36]. Viral DNA was enriched using RCA, as suggested previously by others [40-42]. However it should be noted that RCA may have favorably facilitated the amplification of circular genomes and, as a consequence, hindered the detection of linear genomes. Thus, more than 80% of the classified viral sequences (11,162/13,801) identified in our analysis corresponded to circular viral genomes.
We also agree that more detailed information regarding our sample processing protocol and the corresponding literature references should be provided in the revised manuscript, as follows (Lines 105-114):
Samples were processed according to previously published protocols that have been successfully applied for the identification of papillomavirus (PV) in human skin swab samples [34–36]. Briefly, the cells were centrifuged at 13,000 × g for 5 min and the pellets were resuspended in 100 μl TE buffer (Qiagen, Hilden, Germany), containing 100 μg of proteinase K (Qiagen), and incubated overnight at 55 °C. Following proteinase K inactivation (95 °C for 10 min), the lysates were stored at −20 °C. Subsequently, the obtained samples were tested for the presence of PV DNA using improved versions of FAP [37,38] and CUT PCRs [35], as described previously [36,39]. Circular DNA molecules in lysates of five selected PV-positive samples (four anal and one oral swab) were enriched using rolling-circle amplification (RCA) using the illustra TempliPhi 100 Amplification Kit (GE Healthcare, Chicago, IL, USA) [40–42].
- Table 1 – is it possible to add which viral reads/contigs were detected in the fecal samples and which were detected in the oral samples, or were the oral and fecal samples pooled together?
We appreciate the suggestion of Reviewer 2 to stratify results of read/contig counts by sample type (swab/feces), but, unfortunately, we had to pool all our samples together in order to reach the sufficient nucleic acid concentrations, required by our sequencing provider. Nevertheless the Reviewer’s comment will be taken in consideration for our future studies.
- Table 1 – It strikes me that out of over 10 million read pairs that came out of the analysis, some of the viral families are a single read pair or 1 contig. The authors were very careful to take steps to prevent contamination (and to identify potential contamination, if it happened), and had quality assurance measures in place, but it’s difficult for me to know how to interpret a single (or few) read pairs in such a soup of information. Contigs were >500bp, but how long were the reads, for these instances where there was only a single read? Are you confident those reads are in parts of the genomes that are diagnostic to those viral families? How good were the matches between the reads and the viral families?
Reads were between 130 and 150 bp long. Lines 118-120 in the revised manuscript were modified, as follows:
Sequencing libraries were prepared using the GATC automatic library preparation approach, and the sequencing reads were sequestered in the format of 2 × 150 bp.
Please note that the initial 2×150 bp reads underwent qualitiy trimming of the first and last 15 bp (discussed in lines 120-123 of the revised manuscript).
Indeed, we considered viral families with low read supports (< 20, value we set arbitrarily) as very suspicious. As mentioned in the manuscript we used two different approaches to estimate the virome composition in the samples –1) mapping the reads, and 2) mapping the contigs assembled de novo using those reads (Figure 1). Interestingly in some cases the results of both approaches did not overlap. As Reviewer 2 implies, read-taxon maps at a read length of 150 bp may not hold a high specificity, but such a workflow should be more sensitive than the contig-taxon mapping approach, because many reads may not be incorporated into assemblies, due to being part of the dead-end of the de Brujin graph nodes or due to low edge-coverage.
The version of Centrifuge program we used for taxon-mapping actually outputs listings of taxonomy mappings with a species-level “precision” but did not provide confidence scores for individual sequence-taxon mappings. The last are defined indirectly by setting the program parameters. Due to low likely specificity of short sequence-taxon mappings we decided to back-summarize all the raw centrifuge output results to the level of families (instead of species), to give a more general overview of the viral population and to provide hints where to look for in further studies – which was in fact the aim of our study (as pointed out at lines 72-75 of the current revised manuscript).
Being on the skeptical side ourselves but still aiming to provide a functional starting-point for further studies, we decided that it may be worthwhile to provide a type of confidence-level gradient to our results. First, we are most confident that the two viruses of which we assembled and described the complete genome sequences in detail and annotated them to the species/genotype level, were surely present in the bat samples. Next in terms of the confidence level were the contig-taxonomy mappings, which attempt to replicate the concept of metagenome-assembled genomes (MAGs) in bacterial metagenomics. In contrast to the field of bacterial metagenomics, in viral metagenomics we are often not so lucky to be able to operate with such sequence rich samples. The presence of relatively much smaller genome sizes, and presence of any other, aside of viral, nucleic acids in a given sample, severely hinder the detection and the precision and accuracy of viral characterization population. In our study only 13,897/2,602,700 reads (<1%) (that we know of) originated from viruses; similar studies, such as Geldenhuys et al., 2018 and Dacheux et al., 2014 obtained similar results. Lastly, with the lowest level of confidence we showed also the results of the read-taxon mappings. These should be interpreted with great care, instead of saying “these viruses” are there, our results rather suggest the possibility of the presence of the indicated taxons.
In view of this specific question posed by Reviewer 2 the following paragraph was added to the Discussion section of the revised manuscript (lines 532-555):
Finally, it is worth noting that the results of our metagenomic screening of the pooled T. brasiliensis oral and anal swab samples is effectively provided at three different levels of specificity/sensitivity. The two complete genome sequences (TbraPV1 and TbGkyV1), that have been described at the highest level of detail, also confer the highest level of confidence. In other words, we have complete confidence that these two viruses were present in the pooled nucleic acid samples. On the other hand, we would expect that many viruses that were not assembled to the level of complete genomes were therefore not represented in this group of results. Assigning sequences that did not assemble at the level of complete viral genomes to taxonomical families could therefore be a valid approach offering a higher level of sensitivity than only the complete genome assemblies, but at the cost of diminished specificity. By definition, a great deal of specificity was already lost by summarizing the taxonomical mappings upwards to the family level. In addition, identifying a sequence fragment that resembles a known viral genome in a given genomic region, may not always be sufficient to infer that that specific virus was present in the sample. Viruses are highly promiscuous entities, which can easily exchange parts of their genomes with their hosts, be integrated or even naturalized into the host genomes [107]. Taxonomical viral families identified among the assembled contigs could be interpreted as viral families that were probably represented in our samples. Lastly, and due to the low number of assembled contigs, that were found related to known viral sequences, we attempted to increase the sensitivity even further by obtaining taxonomical family mappings also for the source read pairs. These results, however, should be interpreted with utmost caution, because they likely confer a very low level of specificity due to the limited sequence length (2 × 150 bp). taxonomical viral families identified by read-pair taxonomy mapping only, merely suggest the possibility that these families were present and should be replicated in the future by taxonomical mappings conferring greater specificity, for example with longer sequences, such as those assembled from illumina or nanopore reads.
- Title – because these data represent only 5 bats from a single location, oral/fecal data seem to be combined, and many viral families are represented by only 1 to a few read pairs, I would suggest adding the word “Preliminary” or “Initial” to the beginning of the title.
We agree with Reviewer 2 that our study represents a first approximation of the viruses present in T. brasiliensis. Accordingly, we have changed the title, as follows (lines 2-5):
“A preliminary study of the virome of the South American free-tailed bats (Tadarida brasiliensis) and identification of two novel mammalian viruses”
- For the phylogenetic analyses (Figs 4, 5) – can you provide supplementary information noting the accession numbers and key to abbreviations for the sequences used in these analyses?
The sequences in Figs. 4 & 5 were taken from PaVE (https://pave.niaid.nih.gov/), the reference sequence database for papillomaviruses. According to the suggestion of Reviewer 2 we have added the requested information to revised manuscript, as Table S2. Additionally, we have introduced this information in the Materials and Methods section of the revised manuscript (lines 161-162), as follows:
Additional information on nucleotide sequences (GenBank accession number and virus name abbreviations) used in these analyses is summarized in Table S2.
Moreover, the corresponding heading was included at the Supplementary Material section of the revised manuscript (lines 575-577):
Table S2. List of the 376 reference papillomavirus genomes used for the phylogenetic analyses shown in Figures 4 and 5. Details of viral names, abbreviations, host names and GenBank IDs are provided. Nucleotide sequences were downloaded from PaVe (http://pave.niaid.nih.gov/).
Round 2
Reviewer 1 Report
Thanks to the authors for the revisions - it is more clear, and other studies can access their raw data and contigs from "other viruses".
I have no further queries except perhaps the authors could read through Lines 517-540 again to make the section more concise and check spelling/capitilization.
Thanks for an interesting read.